

# Relationship between climate, environment, and anthropogenic activities in coastal North China recorded by speleothem δ¹⁸O and δ¹³C ratios in the last 1 ka

Qing Wang[1], Ke Cheng[1], Zhihui Zheng[1], Hong Chi[1], Hongyan Wang[1]

[1]*Coast Institute of Ludong University, Yantai 264025, China*

*Correspondence to*: *Qing Wang, schingwang@126.com; Ke Cheng, haian_cheng@126.com*

**Abstract:** Stalagmite KY-1 was collected from Kaiyuan Cave, which is in the warm temperate zone and East Asia monsoon area in Shandong Peninsula, coastal Northern China. Based on the results of U-²³⁰Th-dating, continuous counting of laminae, and linear interpolation/extrapolation, we determined that the stalagmite grew over the last 1 ka between AD 892 and AD
1894. This period includes the Medieval Warm Period (MWP), Little Ice Age (LIA), and an early stage of the Current Warm Period (CWP), and corresponds to a time between the late Tang Dynasty and the late Qing Dynasty of ancient China. We collected 583 samples along the growth axis of stalagmite KY1, and selected 303 samples for δ¹⁸O and δ¹³C ratio measurements according to the Interval test principle to acquire a series of time data. The variation of the δ¹⁸O ratios suggest that a climatic mutation occurred in Shandong Peninsula around AD 1482, corresponding to the MWP/LIA transition period.
The variation of these ratios is generally controlled by the variation of solar radiation. However, the variability of the δ¹⁸O ratios was less than the contemporaneous changes in solar activity in late MWP and late LIA, and more than the contemporaneous changes in solar activity between LIA; the MWP/LIA transition time lagged the contemporaneous solar activity by 30–50 yr. The δ¹³C ratios showed overall synchronization with the δ¹⁸O ratios: both were consistent with the MWP and LIA periods, and both showed the same mutation. However, the curve of the δ¹³C ratios was smoother than that of
the δ¹⁸O ratios, and the variations of both were not synchronized in the late LIA. The transitions between the major dynasties in the past 1000 years of Chinese history, and the late period of Qing Dynasty coincide with the transition periods of the summer monsoon. Furthermore, the impacts of the fluctuation of the sub-climate change trend on the historical evolution of human society in MWP and LIA were more prominent than those of the violent climate transition between MWP and LIA. On the other hand, the degree of land use continually increased in the study area, which, together with the climatic change,
affected the vegetation-soil-groundwater environment system in the mountainous land above the cave. The degree of impact of human land use on the mountainous vegetation, soil characteristics, and their change trends did not exceed the extent of that of climate change from AD 892–1318, but reached or exceeded small climate changes from AD 1318–1497, and reached or exceeded the sharp climate change that occurred between AD 1483–1779. The degree of impact of human land use and the environment stabilized between AD 1779–1894, and the impacts of climate change on the environment were
relatively prominent.



# 1 Introduction

Speleothems are one of the most important terrestrial archives for paleoenvironmental reconstructions (Li et al., 2011). Most previously reported studies of the Asian summer monsoon area are focused on the carbonate oxygen (O) and carbon (C) isotopes, which were used as indicators of the monsoon intensity (Dykoski et al., 2005; Wang et al., 2005; Wang et al.,2008; Liu et al., 2015; Lechleitner et al., 2016). Calcareous speleothems, which have the advantages of precise dating and high-resolution sampling, are becoming one of the best geological archives for dating of major climate shifts and high-resolution reconstruction of past climates and environments (McDermott et al., 2001; Wang et al., 2001; Burns et al., 2003; Genty et al., 2003; Yuan et al., 2004; Zhang et al., 2008; Cheng et al., 2009; Yang et al., 2015; Jamieson et al., 2016). In 2008, we collected the stalagmite KY1 from Kaiyuan Cave in the Shandong Peninsula of northern China, a temperate zone in the East Asia Monsoon area (Zhou et al., 2010). The stalagmite grew over a period of 1 ka from the 9[th] century to the end of the 19[th] century (AD 892–AD 1894). This period includes the Little Ice Age (LIA) and Medieval Warm Period (MWP), both of which had clear records in the history of mankind and strongly influenced human society (Fagan,1999). Base on the U-$^{230}$Th high-precision dating technique and continuous laminae dating, the O and C isotopic ratios of 303 samples of KY1 were measured with an average temporal resolution of ~3 yr. This report is the first example of a high-resolution study of the isotopic values of a stalagmite grown during the last in 1 ka from the temperate zone of the East Asia Monsoon area.

The MWP is considered to range from AD 900–1400 while the LIA ranges from AD 1500–1700 or 1850 (Lamb, 1965; Matthews, et al., 2005; Ogilvie, et al., 2001; National Research Council, 2006). However, the transformation of the two climate periods over the past 1 ka is inconsistent between different areas of the world, and locally distinct climate characteristics and change tendencies are not the same (Broecker, 2001). In Chongqing, dry climates were present during the MWP but wet climatic conditions dominated during the LIA (Li et al., 2011). However, previous investigations indicate that prominent multi-centennial dry periods occurred in Heshang Cave in southwestern China during the LIA (Hu et al., 2008). Paulsen et al., (2003) reported the record of a 1270 yr old stalagmite from the Buddle Cave in the southern Qinling Mountains, which underwent climatic anomalies during the MWP, LIA, and Current Warm Period (CWP), and identified a significant 33 yr climatic cycle that occurred over the last 1270 years. Many scholars have recently researched the climate characteristics and trends of the Asian Monsoon climate area throughout the MWP and LIA. The areas of eastern and northern China influenced by the southeast monsoon are likely to be warm, but not as warm as the areas of southwestern China that are influenced by the southwest Monsoon (Tan, 2007). However, in central China, the MWP climate varied from dry to wet to dry, and was relatively warm (Paulsen, et al., 2003). This pattern not only differs from that of the southwest monsoon area during the MWP(Fleitmann et al, 2003; Wang et al, 2005; Dykoski et al, 2005), but also from that of the areas of northern China that are influenced by the southeast monsoon. Several studies(Tan et al., 2003). Tan et al., (2007) report reconstructions of the temperature of the warm season at Shihua Cave in Beijing, providing evidence of rapid cyclical warming on a centennial scale over the last millennium. Furthermore, a comparison with other high-resolution speleothem





records over the last 750 years indicates regional differences in monsoon precipitation variability between the south and the north of central China on decadal to centennial scales (Tan et al., 2009).

To date, most of the research on the Holocene paleoclimate of the Chinese mainland uses ancient karst stalagmites from the south of the Qinling Mountains-Huai River boundary and the western Qinling Mountains. In the areas north of the Qinling Mountains-Huai River boundary, only Beijing (Shihua Cave), Liaoning (Xiangshui Cave), and other minor regions have been investigated (Tan et al., 2003; Hou et al., 2003). No stalagmite samples suitable for investigation of the climate changes in the period since the MWP have been found in areas north of the Qinling Mountain-Huai River boundary or north of the boundary between the subtropical monsoon and temperate monsoon areas. This deficit is particularly apparent in Shandong peninsula and the adjacent coastal plains of east China, which belong to a typical temperate monsoon area. Furthermore, studies of environmental changes in the Asian Monsoon climate zone have not paid enough attention to the impact of the increase in anthropogenic activities since the MWP on environmental changes; many studies have deduced climate changes directly from the variability of $\delta^{13}$C values, which are controlled both by climate and anthropogenic activities. Therefore, this research follows the work of Wang et al., (2016), using the record of stalagmite KY1 to reconstruct in high resolution the climatic-environmental changes that occurred in the Shandong Peninsula over the last 1 ka. This work will deepen understanding of the characteristics and driving forces of climatic-environmental changes in the temperate Asian monsoon area over the last 1 ka, and will also enrich research on the evolution of the paleoclimate of the East Asian Monsoon Area based on secondary carbonate deposits in karst caves.

## 2 Background

Stalagmite KY1 was deposited in the Kaiyuan Cave (36°24′32″N, 118°02′05″E), located at 175 m above sea level in the northwest hilly area of Lushan Mountain in Zibo city, Shandong province (Fig.1). The Shandong Peninsula is the largest peninsula in China and is located between the Bohai Sea and Yellow Sea. The Middle Cambrian Zhangxia group (consisting of an upper layer of ooid interbedded shale, above a middle layer of shale interbedded with stratum of limestone, oolitic limestone, and algal clot limestone) and the Ordovician of the Badou and Gezhuang groups (the Badou group was named by Chen, (1976)), are widely distributed in its western region. This region is known mainly for the thick gray/dark-gray layer of micrite-thin limestone-interbedded dolomitic limestone and marl, which has a thickness of 24–238 m. The lower section is integrated with the Gezhuang group, and the upper section is disconformity  with the Carboniferous Benxi group (Shandong Provincial Bureau of Geology&Mineral, 1991), which is mainly composed of Lushan Mountain (1108 m), Yishan Mountain (1031 m), and Mengshan Mountain (1156 m). The carbonate rock landforms are well developed in mountainous caves, and many caves are found outcropping on the surface. The well-developed secondary carbonate sedimentary bodies have typical morphological characteristics.

Kaiyuan Cave developed in the dolomite of the Ordovician Zhifangzhuang formation, and the strata have a total thickness of ~110 m. The total length of the cave is 1280 m, and the overall distribution is a northwest–southeast strike with



twists and turns; the width of the space inside the cave is typically 2–8 m and can be up to 30 m. At the top of the cave, the surface of the bedrock is covered by soil with a general thickness of 50–80 cm and up to a maximum thickness of 1.0 m. The soil types are calcareous rocky soil and drab soil (The Soil and Fertilizer Workstation of Shandong Province, 1994), giving priority to broad-leaved forest and xerophytic forest and grass vegetation. The original local vegetation consists of temperate

deciduous and broad-leaved trees, which were anthropogenically deforested, and have been replaced with shrubs and grasses including ziziphus jujuba, bramble, and bothriochloa kuntze. The area around Kaiyuan Cave is currently influenced by both summer and winter monsoons, with an annual precipitation of ~620 mm and an annual mean temperature of ~13 ℃. Summer monsoons prevail during July and August, contributing to half of the annual precipitation (Fig. 2).

## 3 Analytical methods and data processing

The stalagmite KY1 was 75.0 mm in total length and conical in shape, consisting of very pure calcite. No hiatus during its growing process was observed by microscopy on the polished surface or the laminae. The upper part (0–42.769 mm) of KY1 comprised 678 typical transmitting annual laminae overlaid by continuous deposits. We first collected samples from locations at 5 mm, 15 mm, 25 mm, 45 mm, and 64.5 mm, then determined the age using the U-$^{230}$Th high-precision dating technique at the High-Precision Mass Spectrometry and Environment Change Laboratory (HISPEC) of the

National Taiwan University. KY1 had uranium concentrations ranging from 704 to 5147 ppt (Table 1) (Shen et al., 2002). The results of U-$^{230}$Th dating (Table 1) and the chronology established by a linear interpolation between dated points (Fig. 4) indicated that KY1 developed between AD 892 and AD 1894, a period including the MWP and LIA. The growth rate of KY1 varied significantly during this time (Fig. 4).

Since KY1 had no hiatus in its growth, the upper part (0–42.769 mm) was composed of 678 clear and continuous

laminae, and the continuous laminae themselves each had a clear chronology. The timescale of this upper part of KY1 was established by Wang et al., (2016) by counting the annual laminae layer-by-layer to deduce the sedimentation time of each lamina. We counted in the upward and downward directions of this upper part using the laminae that were dated with U-$^{230}$Th technique as markers to first confirm the formation dates of the 1$^{st}$ and 678$^{th}$ laminae, then confirmed the date of each laminae according to its position. In the lower part of KY1 (42.769–75 mm), we used interpolation and extrapolation of the

average growth rate to determine the sedimentation time, and established a timescale based on the U-$^{230}$Th dating results. Then, we calculated the average deposition rate of different laminae according to the age of the 678$^{th}$ laminae and the U-$^{230}$Th dating results of the lower part, and linearly interpolated and extrapolated the age of each sample of the lower part (Fig.4).

KY1 was halved along its growth axis. Firstly, we collected four equally spaced samples, each at 20 mm from the

growth center, along with samples at 9.5 and 18.5 mm from the top and perpendicular to growth axis, respectively, for Hendy tests. Secondly, a 4 ×5 ×75 mm (depth ×width ×length) stone strip was collected along the growth axis of KY1 in its development center. A total of 583 samples were taken from this strip using a medical scalpel to scrape from top to



bottom, with a sampling density of 7–8 samples/mm (average separation distance of 0.1296 mm). Then, 303 samples were selected for $\delta^{18}O$ and $\delta^{13}C$ ratios measurement. We counted the laminae and used interpolation/extrapolation of the average growth rate to determine the sedimentation time of the 303 samples and establish an age model for KY1. Finally, we analyzed the impact of climate and environmental evolution.

Analyses of the stable oxygen and carbon isotope ratios ($\delta^{18}O$ and $\delta^{13}C$ ratios) were conducted along the growth center of KY1 with a temporal resolution of ~3 years. $\delta^{18}O$ and $\delta^{13}C$ isotopes were measured using an automated individual-carbonate reaction (Kiel) device coupled with a Thermo-Fisher MAT 253 mass spectrometer at the State Key Laboratory of Palaeobiology and Stratigraphy of the Nanjing Institute of Geology and Palaeontology, Chinese Academy of Sciences. Each powdered sample (~0.08 to 0.1 mg of carbonate) was reacted with 103% $H_3PO_4$ at 90 ℃ to liberate
sufficient $CO_2$ for isotopic analysis. The standard used was NBS19, and one standard was analyzed with every ten samples. One sample out of ten was duplicated to check the replication. All isotope ratios are reported in permil (‰) deviations relative to the Vienna Peedee Belemnite (VPDB) standard in a conventional manner. The standard deviation ($1\sigma$) for replicate measurements on NBS-19 is < ±0.10‰ for $\delta^{18}O$ and < ±0.03‰ for $\delta^{13}C$.

## 4 Results and discussion

### 4.1 Hendy test results

The $\delta^{18}O$ and $\delta^{13}C$ ratios of the four equally spaced samples and those collected 9.5 and 18.5 mm from the top and perpendicular to growth axis, respectively, indicated that the composition of KY1 was consistent across same positions (Table 2). One sample (KY1-9/10-5) showed a larger $\delta^{13}C$ ratio (−3.368‰) than others in the same position, which had an average value of −4.065‰. The reasons for this variation may be an inaccuracy in the sampling position of KY1-9/10-5, or
that the wrong sample was measured, since the $\delta^{13}C$ ratio (−4.038‰) of another sample (KY1-10-04) collected from the same position was consistent with the average $\delta^{13}C$ value of the other three samples (−4.065‰). The scatterplot of the $\delta^{13}C$–$\delta^{18}O$ ratios of the 303 samples shows no correlation between the $\delta^{13}C$ and $\delta^{18}O$ ratios of KY1 ($R^2 = 0.21$) (Fig.3). Thus, the Hendy test results suggest that the calcite in KY1 was deposited under conditions of isotopic equilibrium. The possibility of dynamic fractionation of the calcite is small in the sedimentary process; therefore, the $\delta^{18}O$ ratio of a stalagmite mainly
reflects the original external climate (Hendy, 1971; Wang et al., 2016).

### 4.2 Timescale of KY1

The upper part of KY1 developed between AD 1217±20 and AD 1894±20 (Wang et al., 2016). In the lower part of KY1, the U-$^{230}$Th dating results of the samples taken at 45 and 64.5 mm were 916.3 ± 39.6 and 1015.1 ± 74.5 yr BP, respectively, corresponding to AD 1096.7 ±39.6 and 997.9 ±74.5 (Table.1). Considering the isotopic dating errors, the date
ranges (AD 1057–1137 and AD 923–1073) of these two sample positions (45 and 64.5mm) overlap, and the ages of the



samples at 75 mm (AD 945) and 64.5 mm (AD 998 ± 75) that were extrapolated from the average deposition rate also overlap. However, if we calculate the average deposition rate according to the ages of the samples at 75 mm (AD 945) and 42.769 mm (AD 1217 ± 20), there is more than a ten-fold difference in the section intervals for the samples at 42.769–45 mm (0.018438 mm yr$^{-1}$) and the section intervals at 45–64.5 mm (0.1969696 mm yr$^{-1}$). Thus, we interpolated the age of the section intervals for samples at 42.769–64.5 mm using the average deposition rate (0.0987772 mm yr$^{-1}$) for this section, and extrapolated the age of the section interval for samples at 64.5–75 mm. This process yielded the age of the laminae at 75 mm (AD 892). Since no hiatus was observed on the polished surface or in the thin section by microscopy, KY1 was assumed to have grown continuously between AD 892 and AD 1894 (Fig. 4).

### 4.3 The variability of the δ$^{18}$O value and influencing factors

The δ$^{18}$O ratios of KY1 ranged from −6.247 to −8.682‰, with average, maximum, and minimum values of −7.8, −6.247 ‰ in AD 1603, and −8.682‰ in AD 930, respectively (Fig. 5A). In the past 1 ka, the δ$^{18}$O ratios showed clear stages of variation; a low value-low fluctuation period followed by periods of high value-high fluctuation period appeared before and after AD 1482, which exhibited a series of centennial to multi-centennial fluctuations including the MWP and LIA. The MWP was indicated by a low value-low fluctuation period (with an average value of −8.0399‰) lasting for 590 years between AD 892 and 1482. In this period, relatively δ$^{18}$O high values were observed in around AD 900, AD 950–1050, and AD 1180–1300, indicating three small climatic fluctuations in the record of stalagmite KY1 during the MWP. The LIA was indicated by a high value-high fluctuation period (average of -7.5980‰) of 403 years, from AD 1482 to AD 1885. In the period from AD 892 to AD 1894, extreme δ$^{18}$O ratios were observed in AD 1498, 1603 (the maximum value), 1687, and 1885, each preceded and succeeded by sharp fluctuations. In AD 1746, the δ$^{18}$O ratios of KY1 returned to its AD 1482 level, but then rose dramatically between AD 1871 and AD 1885.

We compared the variation of the δ$^{18}$O ratios of KY1 (of Kaiyuan Cave) with those from Wanxiang Cave (China, 33°19'N, 105°00'E) (Zhang et al., 2008), located at the edge of the Northern Hemisphere inland temperate monsoon area in Gansu province in China; Sofular Cave (Turkey,41°25'N, 31°56'E) (Fleitmann et al., 2009), located in a Mediterranean climate area of Turkey; Spannagel Cave (Austria,47.0882°N, 11.6715°E) (Mangini et al.,2005), located in a continental climate zone; and UAT Cave (NW Scotland, 58°09'N, 04°59'W) (Baker et al., 2011), located in a temperate maritime climate. The variation of the δ$^{18}$O ratios of these areas all showed a two-stage pattern of low values early in the period, which changed to high values later in the period. While the mutation relation between the low-value period and the high-value period corresponded to the transition of the MWP and the LIA, the δ$^{18}$O mutation time of the stalagmites showed clear differences between different climate zones (Fig. 6). The UAT and Sofular Caves, affected by the maritime westerly wind, showed the latest δ$^{18}$O mutation time; the Wanxiang and Spannagel Caves, nearly unaffected by the ocean, showed the earliest time; and the Kaiyuan Cave, affected by the East Asian Monsoon, fell in between.





Comparing the variation in the KY1 $\delta^{18}$O ratios with the changes in solar activity over the last 1 ka inferred from radionuclide records (Muscheler et al., 2007) indicated that the variations in the KY1 $\delta^{18}$O ratios were synchronized to the changes in contemporaneous solar activity between AD 1000–1430. Both the solar activity and $\delta^{18}$O ratio curves rose strongly between AD 1000–1250 and began to decline from AD 1250 to AD 1430. From AD 1430 to AD 1700, the curve of solar activity raised firstly and then decreased by the boundary of AD 1600, and began to rise in AD 1700 as a whole, the last rising trend ended in AD 1860. However, the curve of $\delta^{18}$O value of the stalagmite KY1 began to rise in AD 1480 and lagged the trend of the solar activity by 50 years. The last rise ended in AD 1885, which lagged solar activity by 25 years, and the highest $\delta^{18}$O value corresponded to the lowest value of solar activity. The peaks and troughs of the solar activity curve corresponded to the $\delta^{18}$O curve of KY1, but the fluctuation ranges of the two curves were different. The fluctuation range of the $\delta^{18}$O ratios was less than that of the contemporaneous changes in the solar activity before AD 1430 and after AD 1700, and increased between AD 1430 and AD 1700 (Fig. 7A).

The relationship between the variation of the $\delta^{18}$O ratios of KY1 and the contemporaneous northern hemisphere (NH) temperature (Esper et al., 2002; Mann et al., 2003; Moberg et al.,2005) was similar to the relationship with the solar activity but with more varied fluctuations in the $\delta^{18}$O value curve. However, the inflection point of the NH temperature curve in AD 1300 lagged the $\delta^{18}$O value curve by 50 years, then by 80 years in AD 1400, and finally by 50 years in AD 1450 (Fig. 7B). Comparing the $\delta^{18}$O value curve to contemporaneous records of Swiss Alpine glaciation (Holzhauser et al., 2005) showed no obvious correlation (Fig. 7C).

The $\delta^{18}$O value of stalagmites in monsoonal areas like eastern China has been used as a proxy for the variability in the amount of rainfall because of the associated changes in the $\delta^{18}$O values with changing moisture sources and shifting rainy season (Yuan et al., 2004). Kaiyuan Cave is located in the warm temperate zone of the East Asia monsoon area. Rainfall is concentrated in the summer months, brought from the low latitudes of the Pacific by the summer monsoon. Thus, changes of the summer monsoon intensity and the amount of precipitation can affect the $\delta^{18}$O value of the stalagmite. We interpreted the $\delta^{18}$O record of KY1 in terms of the Asian summer monsoon intensity and precipitation, as presented by Cheng et al., (2009) and Zhang et al., (2008), respectively: lower and higher $\delta^{18}$O ratios correspond to stronger and weaker summer monsoons, and $\delta^{18}$O ratios correlate inversely with precipitation. Therefore, stronger summer monsoons with more precipitation occurred for 590 years between AD 892–1482, while a weaker summer monsoon with less precipitation occurred for 264 years between AD 1482–1746. A clear change in the summer monsoon intensity and precipitation occurred at AD 1482, corresponding to the transition between MWP and LIA, but lagging the solar radiation intensity by ~50 years (Fig. 5A). The summer monsoon intensity and precipitation decreased from AD 1483, reaching their minima AD 1603. After AD 1603, these values returned to AD 1482 levels, before the MWP/LIA transition in AD 1748. The summer monsoon intensity and precipitation showed obvious increases after AD 1746, but briefly decreased between AD 1871–1885.

Previous research shows that the East Asian summer monsoon intensity depends on the thermal contrast between the Eurasia in the Northern Hemisphere and the low-latitude Pacific Ocean in the Southern Hemisphere (Zhang et al., 2008).



The synchronization between the variation of the δ$^{18}$O ratio and the solar activity indicates that variations in the solar insolation caused this thermal contrast, which lags the solar insolation (Fig. 7A). This is the reason that the changes in the δ$^{18}$O ratio are closely related to the solar insolation but lag its changes by 30-50 years, and are not synchronized with the variation of the temperature in Northern Hemisphere and Swiss Alpine glaciation over the same period, which is also the reason that the δ$^{18}$O ratio variations in the different climate area types of the Northern Hemisphere were not synchronized with the mutation corresponding to the MWP to LIA transition. However, the amplitude of the changes of the KY1 δ$^{18}$O ratio were smaller than those of the changes in solar activity at the end of the MWP (before AD 1430) and at the end of the LIA (after AD 1700), but became larger at the end of MWP and during the LIA. These observations indicate that the East Asia Monsoon climatic system amplified the variation of the solar radiation fluctuation during LIA, and reduced the effect of the changes in solar activity before and after the LIA.

The 1 ka growth period of KY1 covers several ancient Chinese historical periods, including the Tang Dynasty, the Five Dynasties and Ten Kingdoms period, the Northern and Southern Song Dynasty, and the Ming Dynasty before AD 1484, followed by the Ming and Qing Dynasties after AD 1484 (Fig. 5C). The trend of climatic change of the δ$^{18}$O value differences reflected that the summer monsoon intensity showed roughly stronger/weaker alternating changes of the different dynasties above. In the context of these dynasties, the summer monsoon was weak at the end of the Tang Dynasty, then initially strong and later became weak during the Five Dynasties and Ten Kingdoms, then was stronger in the Northern Song Dynasty, weaker in Southern Dynasty, stronger in the Yuan Dynasty, weaker in the Ming Dynasty, and then initially strong followed by weaker and stronger periods in the Qing Dynasty. The transitions between the Tang Dynasty, Five Dynasties and Ten Kingdoms, Northern Song, Southern Song, Ming, Qing, and late Qing Dynasties all coincided with variations in the intensity of the summer monsoon intensity. Thus, the transitions between climate variations, rather than the climate or the variation themselves, were an important natural background in Chinese dynastic history. However, the transition from MWP to LIA was one of the biggest transition in the past 1 ka and occurred during the Ming Dynasty, but did not corresponding with a transition between dynasties. This exception may indicate that the fluctuating changes of the sub-climate variability were influenced by ancient Chinese social and historical evolution during MWP and LIA, which perhaps had a more prominent effect than climate transitions during MWP and LIA.

### 4.4 The variability and influencing factors of KY1 δ$^{13}$C values

Over the last 1 ka of the KY1 record, the δ$^{13}$C ratios ranged from −9.691‰ to −3.095‰, with the maximum value (−3.095‰) appearing in AD 1779, the minimum value (−9.691‰) appearing in AD 1318, and an average value of −7.3208‰ (Fig. 5B). The δ$^{13}$C value records have received extensive attention during this period, and showed clear stages of consistent variation during this period, which can be divided into two sub-stages: low- and high-value. The low-value (average of -8.9329‰) period lasted 590 years, from AD 892–1482, and the high-value period (average of −4.7931‰) lasted 412 years, from AD 1482–1894. The δ$^{13}$C value increased for 297 years during the high-value stage, before reaching its maximum in





AD 1779. This build-up phase was far longer than subsequent decline (115 yr). The $\delta^{13}C$ ratio curve suggests that mutations occurred in AD 1482, and the lower period switched changed rapidly into the subsequent high-value period.

     The $\delta^{13}C$ value depends on the water-rock interaction in the overlying strata of the cave, the paleo-vegetation on the surface including the relative proportions of the C3- versus C4-plants and the bio-productivity. Thus, the variation of the

$\delta^{13}C$ ratio reflects the surface environment of the cave, especially the evolution of the vegetation–soil-groundwater system. The relationship between the $\delta^{13}C$ value and the water-rock interaction (Banner et al., 1996; Baker et al., 1997), the paleovegetation (including the relative proportions of the C3- versus C4-plants) (Holmgren et al., 1995; Dorale et al., 1998), and the bio-productivity (Genty et al., 2003) is such that higher proportions of C3- versus C4-plants, increased bio-productivity, and lower water-rock interaction all contribute to lower $\delta^{13}C$ ratios, and vice versa. The $\delta^{13}C$ ratios of KY1

indicate a lower proportion of C3- versus C4-plants, lower bio-productivity, and less water-rock interaction before AD 1482. After this time, a mutation to higher proportions of C3- versus C4-plants, higher bio-productivity, and more water-rock interaction led to $\delta^{13}C$ values reaching a maximum in AD 1779. The $\delta^{13}C$ value then began to decline, but remained much higher than the AD 1482 level until KY1 stopped growing.

     There were many similarities between the $\delta^{18}O$ and $\delta^{13}C$ records of KY1, but there were also several important

differences. Firstly, both $\delta^{18}O$ and $\delta^{13}C$ records showed obvious mutation phenomena, and most of the climatic fluctuations present in the $\delta^{18}O$ records were also evident in the $\delta^{13}C$ record. The $\delta^{13}C$ records showed that changes in the proportions of C3- versus C4-plants, bio-productivity, and water-rock interactions occurred in AD 1482 (Fig.5 A, B), which corresponded closely to the changes from strong to weak summer monsoon intensity and precipitation in AD 1482 shown in the $\delta^{18}O$ records. This mutation involved changing from a low-value period to a high-value period, but the degree of fluctuation in the

$\delta^{13}C$ ratios was less than that in the $\delta^{18}O$ records. The $\delta^{13}C$ ratio curve was also smoother (Fig.5A, B). Furthermore, larger differences between the $\delta^{13}C$ and the $\delta^{18}O$ variations occurred later; the $\delta^{18}O$ ratio reached its highest value in AD 1603 while the $\delta^{13}C$ ratios did not reached its highest point until AD 1779, a delay of 176 years (Fig. 5A, B). On the other hand, both variation tendencies showed synchronization with each other before AD 1603. However, the $\delta^{18}O$ ratios decreased from AD 1603, and reached the former level of AD 1482 before the mutation of AD 1746 and decreased unceasingly. The $\delta^{18}O$

ratios had a short increasing stage since AD 1871 and reached the former mutation levels again in AD 1885, but the $\delta^{13}C$ ratios variations were out of sync with the $\delta^{18}O$ ratios and still much higher than the former level before AD 1482 until the stalagmite stopped growing of AD 1894.

     The variations of the surface vegetation-soil-groundwater system were reflected by the relative proportions of the C3- versus C4-plants, bio-productivity, and water-rock interaction, which are also related to both local climatic changes and

anthropogenic activities. These activities include utilization of land on the mountain surrounding the cave, as the associated disafforestation, reclamation of cultivated land, and growth of crops affects the physical and chemical properties of the soil and changes the proportion of C3- versus C4-plants. The $\delta^{13}C$ ratios were synchronized with the $\delta^{18}O$ ratios from the late Tang Dynasty in AD 892 to the middle of the Yuan Dynasty in AD 1318 but fluctuated less than the $\delta^{18}O$ ratios, which





indicates that anthropogenic activities had a significant impact on the surface environment but did not change the composition, characteristics, or variation of the natural vegetation.

From AD 1318 to the mid-Ming Dynasty in AD 1479, the $\delta^{13}C$ ratios varied as the opposite of the $\delta^{18}O$ ratio; the $\delta^{18}O$ and $\delta^{13}C$ ratios tended to decrease and increase, respectively, in a series of small fluctuations. Thus, we infer that the impact of anthropogenic activities reached or exceeded those of small climatic fluctuations, changing the natural surface vegetation composition, soil characteristics, and variation tendencies. While the $\delta^{13}C$ ratio increased from AD 1483 to the mid-Qing Dynasty in AD 1779, the $\delta^{18}O$ ratios consistently returned to AD 1483 levels after reaching peaks in AD 1498, 1603, and 1687. The $\delta^{13}C$ curve only showed small fluctuations in response to the $\delta^{18}O$ curve. During this period, the population increased rapidly, with consequent increases in the demand for firewood and the cultivation of sweet potato, peanut, and corn; the cultivated area of herbaceous plants continuously expanded. Kaiyuan Cave is in Shandong province, where the cultivated area increased from 54292938 mu in AD 1502 to 9934263 mu in AD 1753 (Liang, 1980) (Fig. 5c). Since the plains and lower-lying areas had been completely developed and utilized, reclamation of mountainous areas provided the main sources of new land for cultivation. Crops became the main surface cover instead of the original shrubs and trees, leading to the intensity of the impact of anthropogenic activities on the vegetation and soil of the mountain reaching and exceeding those of the climate. The $\delta^{13}C$ and $\delta^{18}O$ ratios decreased between AD 1779–1894, except for a period of increase after AD 1885, because the cultivated land area and the land utilization were relatively stable over this period (98634511, 98472844, and 98472846 mu in AD 1812, 1851, and 1873, respectively) (Fig.5c). While the impact of anthropogenic activities on the vegetation-soil stabilized, the variation of the $\delta^{13}C$ value remained prominent under the impact of the overall enhancement of the summer monsoon at the end of the LIA.

The drought/waterlog index cumulative departure curve (Chinese Academy of Meteorological Sciences of China Meteorological Administration, 1981; Wang et al., 2016), $\delta^{18}O$ ratios, and $\delta^{13}C$ ratios have the same variation tendencies (Fig. 5). In the cumulative departure curve, an increase corresponds to the changes associated with a drier environment and a declining trend corresponds to the changes associated with becoming waterlogged. The curve shows that precipitation decreased from AD 1480, which corresponds to the MWP to LIA transition period. At this time, $\delta^{18}O$ ratios show a mutation from a low-value low-fluctuation period to a high-value high-fluctuation period, and the $\delta^{13}C$ ratios change from a low-value period to a high-value period. The cumulative departure curve shows a period of lower precipitation from AD 1480–1744, which corresponds to the high-value high-fluctuation period in the $\delta^{18}O$ ratios curve and indicates that, in the weaker summer monsoon period, the degree of fluctuation of the summer monsoon intensity increased and the precipitation decreased. The cumulative departure curve shows a lower precipitation period from AD 1480–1744 that coincides with a fast rise in the $\delta^{13}C$ ratios; these effects may indicate that the crop yield of cultivated land decreased because of drought from the low precipitation period. In response, humans may have been forced to expand the area of cultivated land by reclaiming wastelands (e.g. forested and mountainous areas) to obtain enough food to survive. This behavior results in higher proportions of C3- versus C4-plants and increased bio-productivity. Furthermore, a period of higher precipitation after ~AD





1744 corresponded to a reduction in the $\delta^{13}$C ratios. Thus, we conclude that anthropogenic activities affect the variation of $\delta^{13}$C ratios.

**5 Conclusions**

This study presents the first high-precision dating and high-resolution $\delta^{18}$O and $\delta^{13}$C records of the climatic-environmental changes in the Asian temperate coastal monsoon area over the past 1 ka, based on measurements of a stalagmite, KY1, from Kaiyuan Cave in the Shandong Peninsula. The results of the U-$^{230}$Th dating, continuous laminae counting, and measuring the $\delta^{18}$O ratios and $\delta^{13}$C ratios (average temporal resolution of ~3 yr) of the stalagmite KY1 showed a series of centennial to multi-centennial climate fluctuations that were broadly similar to those documented in Northern Hemisphere calcareous speleothems, including the MWP and LIA. The $\delta^{18}$O ratios of KY1 show that the Shandong Peninsula underwent a climate mutation in ~AD 1482, before which there was a strong summer monsoon and more precipitation than after AD 1482 (the weakest summer monsoon and the least precipitation appeared in AD 1609). In addition to the MWP to LIA climate transition, which occurred during the Ming dynasty, many other short-term secondary climate fluctuations have occurred in the past 1 ka, and these fluctuations synchronize with transitions between ruling dynasties.

When compared alongside the changes in solar insolation over the same period, the variations in the $\delta^{18}$O ratios of KY1 were found to correspond to the variations in solar radiation. The ranges of variation of the $\delta^{18}$O ratios was less than that of the changes in solar activity in the later MWP and the later LIA after AD 1603, and more than in the same period of solar activity changes in the high LIA and between the two periods. The conversion of the MWP/LIA lagged by 30–50 yr compared to the solar activity in the same period. The $\delta^{18}$O value changes of KY1 were compared to Alpine glacial records and NH temperature reconstructions during the same stage, but were not found to be synchronized with either series. The $\delta^{18}$O records of KY1 showed low to high trends and mutations corresponding to the MWP and LIA transition that were similar to those of other stalagmite records from the same time. These records were influenced to differing degrees by the impact of the ocean on climate zones in the Northern Hemisphere, but there were clear differences between the mutation time and the climate change process during the two periods. These results show that the Shandong peninsula in Northern China, located in a warm temperate zone and monsoon area, responded to the global climate change in a different way to other areas, and suggest that the climate system response to the solar radiation corresponded to the degree of impact of the ocean and influenced the MWP/LIA shift.

According to the $\delta^{13}$C record of KY1, an abrupt change to the proportions of C3- versus C4-plants, bio-productivity, and water-rock interaction occurred in the area of Kaiyuan Cave occurred in AD 1482: higher proportions of C3- versus C4-plants, increased bio-productivity, and lower water-rock interaction all contributed to lower $\delta^{13}$C ratios before AD 1482; lower proportions of C3- versus C4-plants, decreased bio-productivity, and higher water-rock interaction all contributed to





higher $\delta^{13}C$ ratios after AD 1482. These factors reversed after AD 1482 and reached maxima in AD 1779 before decreasing again, though the levels of each factor at the end of KY1's growth period in AD 1894 remained higher than those of AD 1482. The variations of the $\delta^{13}C$ ratios of KY1 were not synchronized with the $\delta^{18}O$ ratios; both were composed of MWP and LIA, but the curve of the $\delta^{13}C$ ratios was smoother. The variations of the $\delta^{13}C$ and $\delta^{18}O$ ratios were out of sync in the

LIA, and were more prominently unsynchronized later in the LIA. The lack of synchronization between the climatic changes and environmental variations was indicated by the proportions of the C3- versus C4-plants, bio-productivity, and water-rock interaction variations shown by the $\delta^{13}C$ record of KY1. The $\delta^{13}C$ record was not only associated with the regional summer monsoon intensity and precipitation variation, but also with the intensity of human land utilization, which continuously increased in the mountainous area around the Kaiyuan cave from the end of Tang Dynasty to the end of Qing

Dynasty. Anthropogenic activities increased especially rapidly in the 17th century, corresponding to the early Qing dynasty, such that the degree of impact of anthropogenic activities on the vegetation and soil around the cave exceeded that of the climate. This increase resulted in a reverse of the $\delta^{13}C$ variation tendency of KY1 compared to the climate variation tendency shown by the $\delta^{18}O$ ratios.

      The temporal relationship between anthropogenic activities, the climate, and the environment coincides with the

transition periods between the majority dynasties in Chinese history, and particularly the late Qing Dynasty. In the past 1 ka, these dynastic changes coincided with transitions between different summer monsoon stages. Therefore, the transitions of climatic variation tendency were an important natural background to the dynastic transitions of Chinese history. Furthermore, the impact of the trends in the fluctuations of climatic variation tendency in the MWP and LIA on the historical evolution of human society was shown to be more prominent than the MWP to LIA climate transition itself. On

the other hand, the degree of impact degree of human land utilization on the mountain vegetation composition, soil characteristics, and the trends in their changes were less significant than climate change from AD 892 to AD 1318, but the effects of human land utilization reached or exceeded those of small climate changes from AD 1318 to AD 1497. These effects significantly exceeded climate change between AD 1483–1779, but the impact of human land utilization and the environment tended to be stable from AD 1779–1894. Thus, the impacts of climate change on the environment were

relatively prominent.

**Acknowledgement**

      This research was funded by the National Natural Science Foundation of China (NNSFC, NO. 41171158). The authors thank Dr. Shen Chuan-Chou(National Taiwan University) and Jiang Xiuyang (Fujian Normal University) for his help in high-precision dating with the U–Th techniques, and thank Houyun Zhou (South China Normal University)

in sample collection and analysis.





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



**Figures**

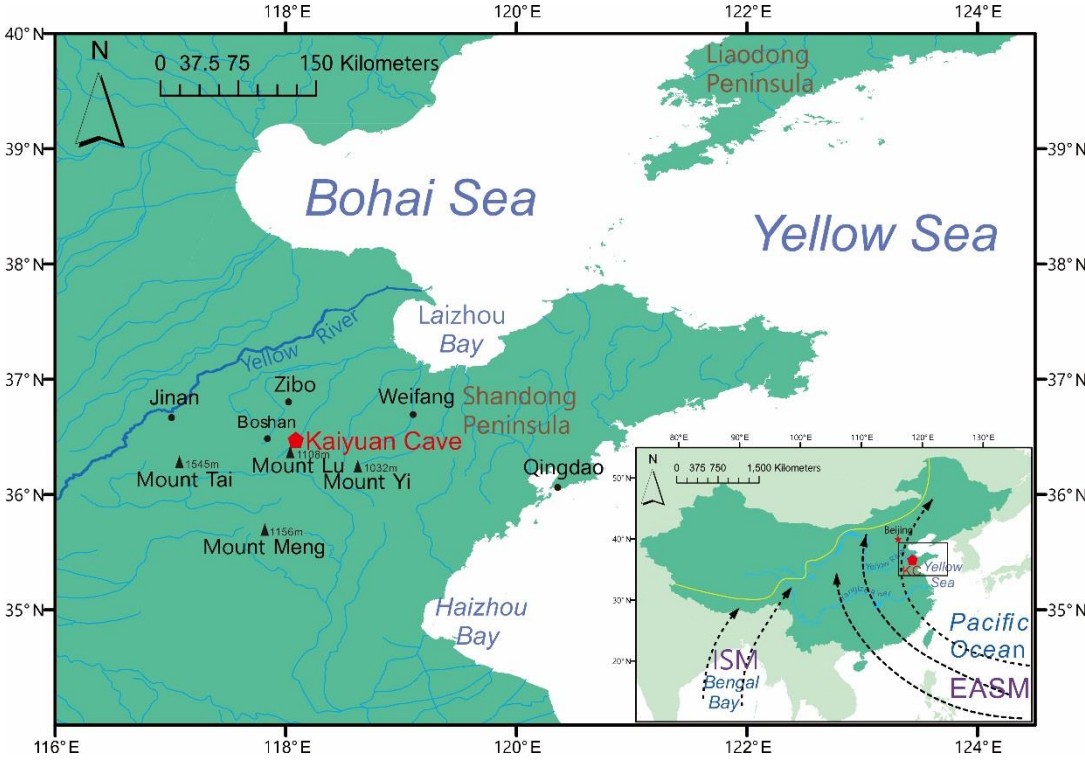

**Figure 1 Location of Kaiyuan Cave and Shandong peninsula in monsoonal China. KC: Kaiyuan Cave (36°24′32″N, 118°02′05″E).**
5   **The thick blue line is the Yellow River, and the thin blue lines are other rivers in this area. In the bottom-right inset, the yellow line indicates the northwestern boundary of the Asian summer monsoon. The dashed black lines with arrows indicate the routes of the summer monsoon. EASM: East Asia Summer Monsoon. ISM: Indian Summer Monsoon.**



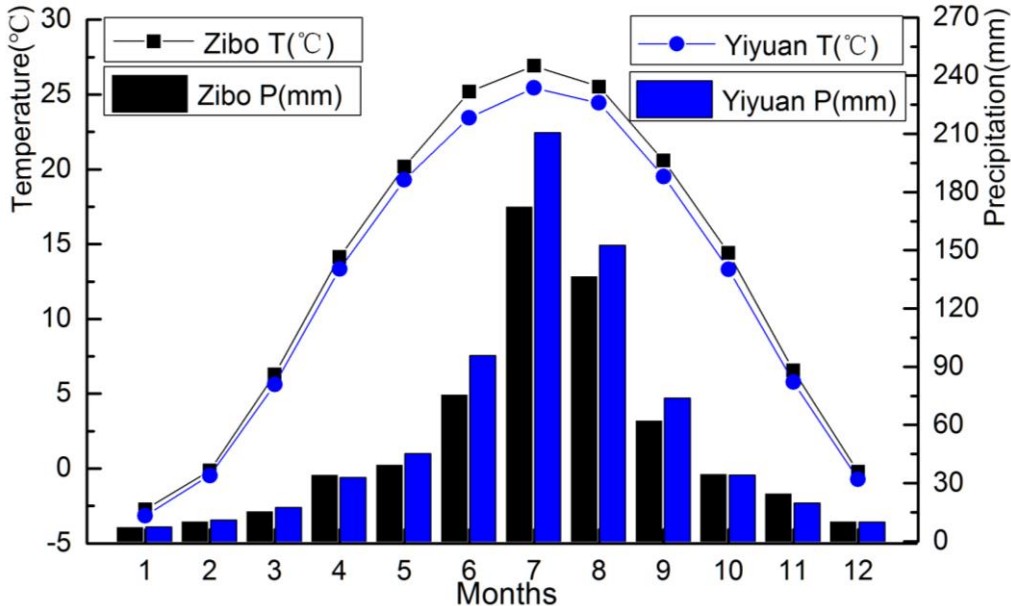

**Figure 2 Monthly mean temperature (T) and precipitation (P) at Zibo Station (1956-1994) and Yiyuan Station (1958-2005), the two meteorological stations close to the study site.**

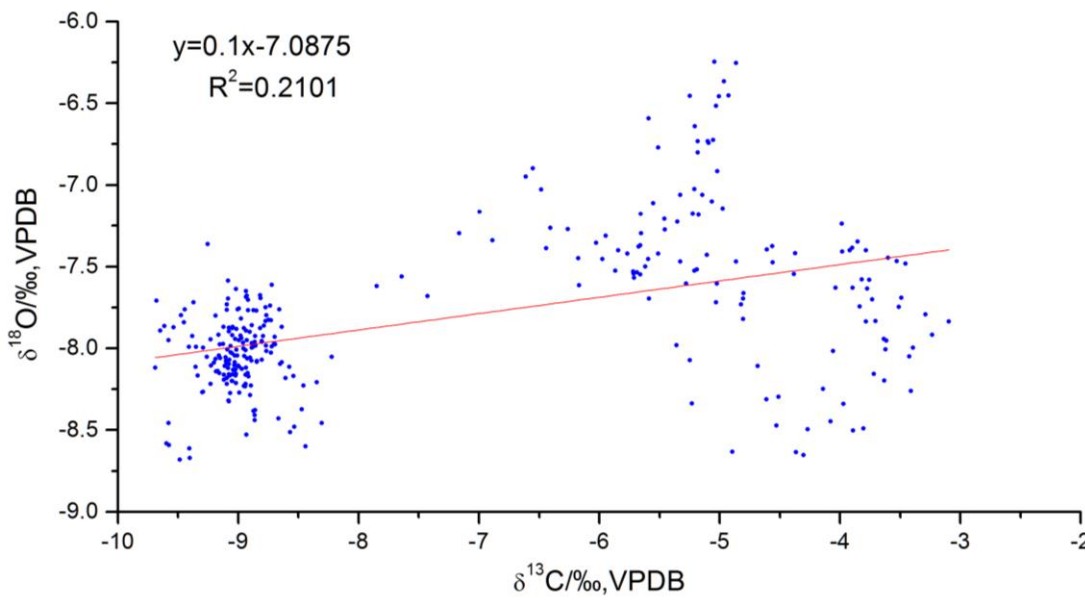

**Figure 3 Correlation between δ¹⁸O and δ¹³C values in the stalagmite KY1, which indicate that the calcite in KY1 was deposited under isotopic equilibrium conditions.**



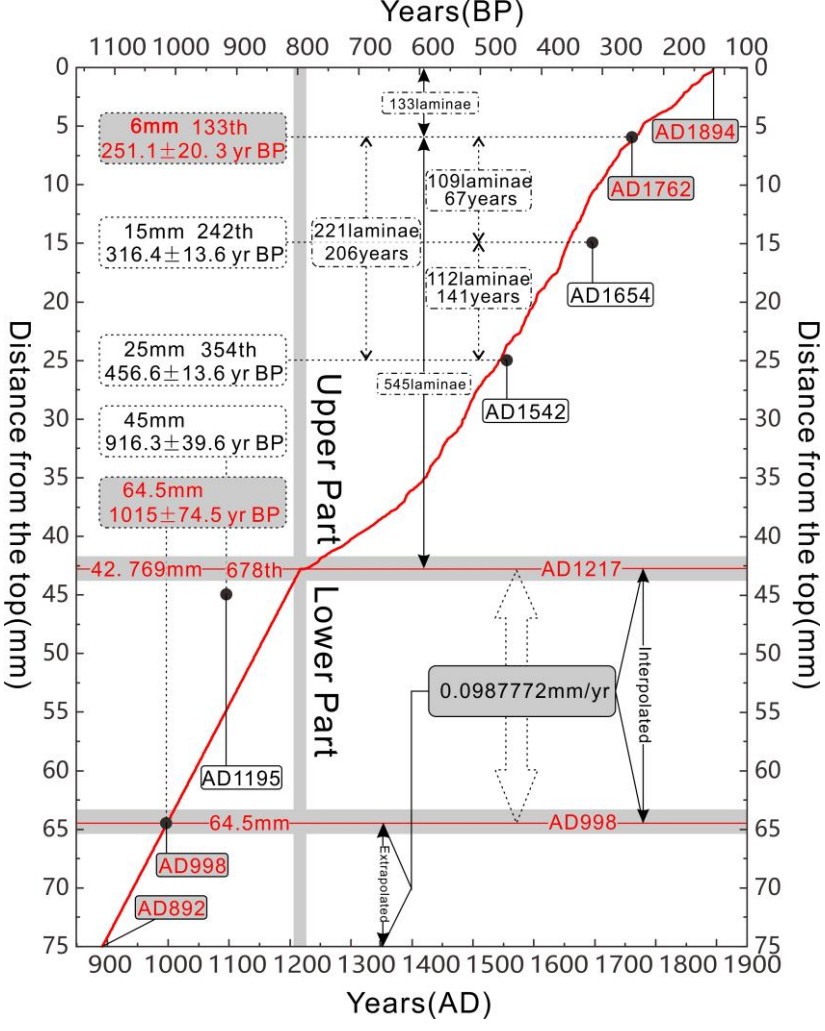

**Figure 4 Age model for stalagmite KY1 established by a combination of laminae counting and linear interpolation/extrapolation based on the average growth rate at dated points. The black points indicate dated points based on high precision dating results using U-230Thdating. The red line represents the time scale of KY1. The upper part (0–42.769 mm) of stalagmite KY1 was calculated by the horizon of 6 mm (133rd laminae) by continuous laminae counting upwards and downwards. The lower part (42.769–75 mm) of stalagmite KY1 was calculated from the age of the 42.769 mm (AD 1217) and 64.5 mm (AD 998) samples based on linear interpolation (42.769–64.5 mm) or extrapolation (64.5—75 mm) of the average deposition rate (0.0987772 mm yr⁻¹). "BP (before present)," "present" in this table refers to AD 2013.**



**Figure 5 Comparison of the δ¹⁸O and δ¹³C records of KY1 from Kaiyuan cave in the Shandong peninsula and the cultivated land area of Shandong province from AD 1500 to AD 1890. The gray bands indicate the mutation points of the δ¹⁸O and δ¹³C values. (A) The δ¹⁸O value time series of KY1, including the MWP and LIA, in which the arrow indicates that the transition from the MWP to LIA at AD 1482. The maximum δ¹⁸O value of −6.247‰ appeared in AD 1603 and the minimum of −8.682‰ in AD 930. (B) The δ¹³C value time series, with a mutation point at AD 1482, maximum value of −3.095‰ at AD 1779, and the minimum value of −9.691‰ at AD 1318. (C) The black line shows the area of cultivated land in the Shandong province from AD 1500–1890; the orange line is the cumulative departure curve of drought/waterlog index. In the figure, ancient Chinese dynasties are indicated with 5D & 10K standing for the Five Dynasties and Ten Kingdoms period.**





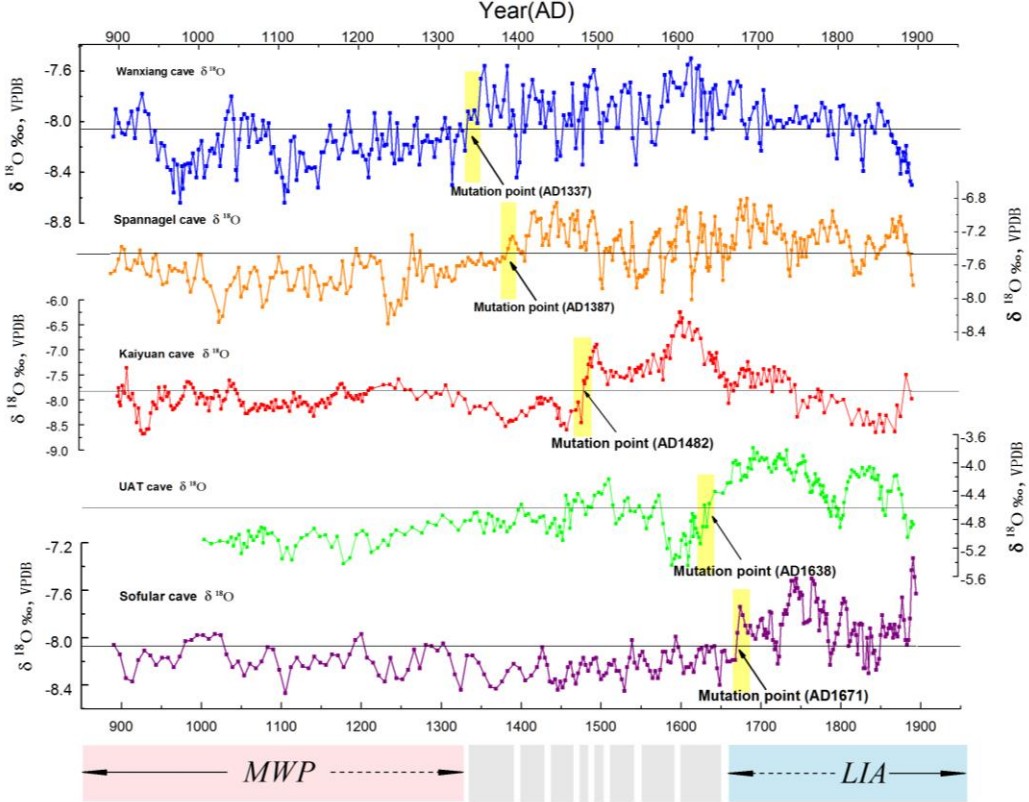

**Figure 6 δ¹⁸O value time series of the stalagmites collected from the Wanxiang Cave (Zhang, 2008) (blue line), the Spannagel Cave (Mangini et al., 2005) (orange line), the Kaiyuan Cave (red line), the UAT Cave (Baker, 2011) (green line), the Sofular Cave (Fleitmann et al., 2009) (purple line) in the last 1 ka, and their relationships. Black lines mean the average values of the δ¹⁸O of 5 every cave. The mutation points of every stalagmite in the last 1 ka were AD 1337 for the Wanxiang cave, AD 1387 for the Spannagel cave, AD 1482 for the Kaiyuan cave, AD 1638 for the UAT cave, and AD 1671 for the Sofular Cave respectively.**





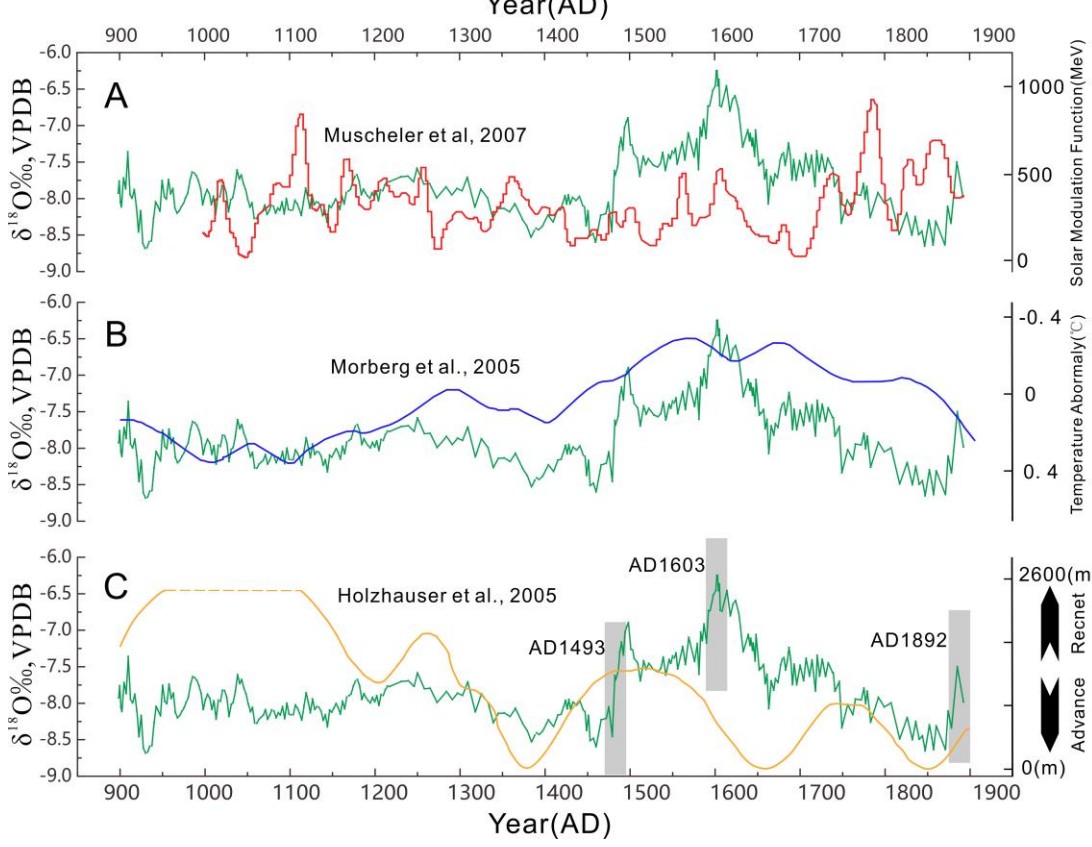

**Figure 7** Comparisons of the KY1 δ18O values, solar irradiance changes, NH temperature reconstructions and Alpine glacial records. The δ18O time series line of KY1 is in green. (A) Solar irradiance from 10Be records (Muscheler et al., 2007) (B) NH temperature reconstructions (Morberg et al., 2005). (C) Swiss Alpine glaciation (Gorner glacier) (Holzhauser et al., 2005).





**Tables**

**Table 1** U-series isotopic results and ages for stalagmite KY1 from Kaiyuan Cave, Shandong Peninsula, Northern China.

| Sample ID | 1 | 2 | 3 | 4 | 5 |
|---|---|---|---|---|---|
| Distance from top (mm) | 6.0 | 15.0 | 25.0 | 45 | 64.5 |
| $^{238}$U ppb[a] | 347.47± 0.63 | 434.45± 0.92 | 334.58± 0.61 | 374.63± 0.73 | 379.91± 0.61 |
| $^{232}$Th ppt | 1245.2± 5.0 | 959.9± 4.9 | 704.6± 5.1 | 2514.6± 9.3 | 5147.0± 19.0 |
| $\delta^{234}$Umeasured | 1457.9± 5.5 | 1341.2± 5.1 | 1320.3± 4.6 | 1468.7± 5.1 | 1522.8± 4.9 |
| $[^{230}$Th/$^{238}$U] activity[c] | 0.00652±0.00014 | 0.00732±0.00011 | 0.01021±0.00013 | 0.02229±0.00037 | 0.02665±0.00051 |
| $[^{230}$Th/$^{232}$Th] ppm[d] | 30.0± 0.68 | 54.63± 0.89 | 79.9± 1.2 | 54.75± 0.94 | 32.42± 0.63 |
| Age uncorrected BP | 289.6± 6.5 | 341.4± 5.4 | 480.6± 6.3 | 988.2± 16.8 | 1157.1± 22.5 |
| Age corrected[c,e] BP | 251.1± 20.3 | 316.4± 13.6 | 456.6± 13.6 | 916.3± 39.6 | 1015.1± 74.5 |
| Age corrected[c,e] AD | 1761.9±20.3 | 1696.6±13.6 | 1556.4±13.6 | 1096.7±39.6 | 997.9±74.5 |
| $\delta^{234}$Uinitial corrected[b] BP | 1458.9± 5.5 | 1342.4± 5.1 | 1322.1± 4.6 | 1472.5± 5.2 | 1527.2± 4.9 |

Chemistry was performed on July 8, 2013 with the analysis method of Shen et al., (2003), and instrumental analysis on Multi Collector Inductively Coupled Plasma Mass Spectrometer (MC-ICP-MS) (Shen et al., 2012). Analytical errors are 2σ of the mean. "BP (before
5  present)," "present" in this table refers to AD 2013.

[a]$[^{238}$U] = $[^{235}$U] × 137.818 (±0.65‰) (Hiess et al., 2012); $\delta^{234}$U = ($[^{234}$U/$^{238}$U]$_{activity}$ - 1) ×1000.

[b]$\delta^{234}$U$_{initial}$ corrected was calculated based on $^{230}$Th age (T), i.e., $\delta^{234}$U$_{initial}$ = $\delta^{234}$U$_{measured}X$ e$^{\lambda 234*T}$, and T is corrected age.

[c]$[^{230}$Th/$^{238}$U]$_{activity}$ = 1 - e$^{-\lambda 230T}$ + ($\delta^{234}$U$_{measured}$/1000)[ $\lambda_{230}$/($\lambda_{230}$ - $\lambda_{234}$)](1 - e$^{-(\lambda 230 - \lambda 234)\,T}$), where $T$ is the age. Decay constants are 9.1705 × $10^{-6}$ yr$^{-1}$ for $^{230}$Th, 2.8221 ×$10^{-6}$ yr$^{-1}$for $^{234}$U (Cheng et al., 2013), and 1.55125 ×$10^{-10}$ yr$^{-1}$ for $^{238}$U (Jaffey et al., 1971).

10  [d]The degree of detrital $^{230}$Th contamination is indicated by the $[^{230}$Th/$^{232}$Th] atomic ratio instead of the activity ratio.

[e]Age corrections for samples were calculated using an estimated atomic $^{230}$Th/$^{232}$Th ratio of 4 ±2 ppm. Those are the values for a material at secular equilibrium, with the crustal $^{232}$Th/$^{238}$Uvalue of 3.8. The errors are arbitrarily assumed to be 50%.



**Table 2 The results of the Hendy tests conducted along two growth laminae of KY1 at depths of 9.5 mm and 18.5 mm individually, which indicate that calcite in KY1 was deposited under isotopic equilibrium conditions according to the Hendy Test rules(Hendy, 1971; Wang et al., 2016).**

| Sample number | Distance from the top/mm | Distance from the center of growth/mm | $\delta^{13}C_{PDB}$ | $\delta^{18}O_{PDB}$ |
|---|---|---|---|---|
| KY1-9/10-5 | | 5.0 | -3.368‰ | -7.506‰ |
| KY1-9/10-10 | | 10.0 | -4.125‰ | -7.753‰ |
| KY1-9/10-15 | 9.5 | 15.0 | -4.096‰ | -7.981‰ |
| KY1-9/10-20 | | 20.0 | -3.975‰ | -7.691‰ |
| KY1-18/19-5 | | 5.0 | -5.138‰ | -6.571‰ |
| KY1-18/19-10 | | 10.0 | -5.301‰ | -6.671‰ |
| KY1-18/19-15 | 18.5 | 15.0 | -5.260‰ | -6.540‰ |
| KY1-18/19-20 | | 20.0 | -5.229‰ | -6.542‰ |

