# Peer review of "Relationship between climate, environment, and anthropogenic activities in coastal North China recorded by speleothem $\delta^{18}O$ and $\delta^{13}C$ ratios in the last 1 ka"

_Climate of the Past, 2017_

## Referee Comment (RC1) · Anonymous Referee #1 · 28 Aug 2017

The manuscript entitled "Relationship between climate, environment, and anthropogenic activities in coastal North China recorded by speleothem $\delta$18O and $\delta$13C ratios in the last 1 ka" by Wang et al. extends the previous Kaiyuan cave record (Wang et al., 2015 Marine Geology and & Quaternary Geology; Wang et al., 2016 Clim. Past) from $\sim$AD1200 further back to AD 900. While the $\delta$13C record is new, the $\delta$18O is essentially the same as the data published previously. The majority of the discussions/conclusions is not only tentative and/or ambiguous (see examples as listed below), but also already published in Wang et al., 2015 Marine Geology and & Quaternary Geology and Wang

et al., 2016 Clim. Past. As such, this manuscript is not suitable for considering publication in Climate of the Past.

Overall, the manuscript has no significant new contributions. Additional comments are listed below:

1. More than half of the abstract is virtually as same as those in Wang et al., 2015 Marine Geology and & Quaternary Geology, and Wang et al., 2016 Clim. Past. 2. The link between the Kaiyuan record and Chinese cultural history is not convincing. For instance, if the Kaiyuan record is indeed a rainfall amount proxy on large spatial-scale in China, how about the differences with other existing records (such as Wangxiang, Heshang and Shihua records)? It really requires a detailed discussion how a record from 'the warm temperate zone (also need a definition)' can affect hydrological condition in China and thus the Chinese culture history. 3. It is necessary to give the reasoning why the 230Th age at $\sim$ 45mm was discarded. 4. The extended portion of the record has very poor age control and the methodology is problematic (e.g., the assumption of linear-growth is too weak). Thus, the new record cannot be used to address the issues in the way that presented in the current manuscript. 5. The age uncertainties are not carefully considered throughout the manuscript when discussing relent issues such as age comparison, and the lead/lag among climate forcings. For example, the Five Dynasties and Ten Kingdoms has a duration less than the age uncertainty of the cave record at the time, and thus their correlation in the Figure 5 needs a justification. 6. The authors interpreted the $\delta$13C record as an indicator of the land use. Given the fact of significant correlation between the $\delta$13C and $\delta$18O data (r=0.46, p<0.01), what about the $\delta$18O? Any anthropogenic (e.g., land use) effect? The data of land use are an overall summary from Shandong Province, which are not necessary to be equivalent to or describe the local variations at the cave site. 7. The authors had published the "Hendy Test" data already. In addition, the simple test presented in the manuscript is not necessary to be a robust verification of 'sample deposition under isotopic equilibrium'. 8. The statement, "This report is the first example of a high-resolution study", is

[Figure]

not proper, regarding many existing records, including that in authors' last paper (Wang et al., 2016 Clim. Past). 9. The $\delta$18O variation is causally linked to the rainfall amount effect. This requires a very careful assessment. 10. The reinterpretation of other cave records in the manuscript is problematic. For instance, the Wangxiang record is also an East Asian monsoon record, rather than a typical Westerlies record. 11. Many climate records are now available for the last millennia in the East Asian monsoon region. I suggest considering a more comprehensive comparison. The comparison with records from Turkey and Europe is ambiguous and not helpful here, unless the authors provide a mechanism to explain their correlations.

13. Almost all reported data in the manuscript have too many significantdigits, which is obviously impossible. 14. The conclusion part is unusually long with many redundant contents. 15. The current manuscript is not sufficiently comprehensible, including English. 16. Some references are not very appropriate and some need to update. 17. Check the unit of U contents: ppt or ppb?

Please also note the supplement to this comment:
https://www.clim-past-discuss.net/cp-2017-73/cp-2017-73-RC1-supplement.pdf

---

## Referee Comment (RC2) · Anonymous Referee #2 · 15 Sep 2017

In their manuscript, Wang et al. tried to use oxygen and carbon stable isotope records in a stalagmite sample from Kaiyuan Cave to address the relationship between climate, environment and human activities. However, the manuscript was very poorly written and some parts are even unreadable (e.g., lines 25-30 in abstract and lines 23-25 in section 4.3). There are numerous grammar mistakes and redundancies in the writing (e.g., the first two paragraphs in section 3). Moreover, the authors often made awkward statements without reasoning. It is difficult for me to recommend its publication in Climate of the Past. Following I provide a few major comments and minor issues, and

hope they would be helpful in authors' resubmission and future research.

Major issues: 1. I am surprised that the current manuscript has a significant overlap with their previous paper published in Climate of the Past (Wang et al., 2016, 12, 871-881), although the authors did include new datasets (e.g., 2 more U/Th dates, d13C data) in this manuscript. Some paragraphs were even simply copied from the previous one (e.g., in introduction, background, . . .). This is typically unacceptable in scientific journals.

2. The age model the authors constructed for the sample is not reliable. How could it be possible to assign a specific calendar year to a laminar, particularly given the slow growth rate and relatively large U/Th age error bar? Some U/Th dates were randomly thrown away, without careful reasoning.

3. The authors really need to learn how to scientifically present their data. They need to add errors when present measurement data. Significant figures also need to be considered.

4. Section 4, it would be beneficial to show pictures of sample KY1, its lamina and micro-sampling locations.

5. In section 4.1, why not re-measure the subsamples if the authenticity of the sample is uncertain?

6. In section 4.2, it seems very odd to exclude the U-Th age at 45mm from the age model.

7. In section 4.3, the authors observe a quite interesting feature that "The UAT and Sofular Caves, affected by the maritime westerly wind, showed the latest $\delta18O$ mutation time; the Wanxiang and Spannagel Caves, nearly unaffected by the ocean, showed the earliest time; and the Kaiyuan Cave, affected by the East Asian Monsoon, fell in between." What's the possible mechanism behind this phenomenon?

8. In section 4.3, the relationship between solar activity and $\delta18O$ were not sufficiently

discussed. The authors argued changes in $\delta18O$ are corresponding to the variations in solar radiation (although I would argue it is not visually convincing). However, the authors did not explain why the changes of $\delta18O$ lags the changes of solar radiation and why the changes of $\delta18O$ respond to solar changes differently before, during and after LIA. These statements are very subjective.

9. In section 4.4, the authors argue that the variations of speleothem $\delta13C$ is controlled by the changes of proportion of C3- vs C4-, bio-productivity and the water-rock interaction. First, how to quantify the water-rock interaction? by time? Second, why the low $\delta13C$ values before 1482AD necessarily "indicate a lower proportion of C3- versus C4-plants, lower bio-productivity, and less water-rock interaction" (page 9 line 10)? Third, "This behavior results in higher proportions of C3- versus C4-plants and increased bio-productivity." (page 10 line 33), which according to the authors will decrease the speleothem $\delta13C$ values, contradicting to the increase trend of $\delta13C$ values between 1480-1744 AD.

10. In section 4.3 and 4.4, the relationship between the ancient Chinese dynasties and stable isotopes are very weak.

Minor issues:

Page 1 line 19, what does "smoother" mean here?

Page 2 line 25, the authors stated "The areas of eastern and northern China influenced by the southeast monsoon are likely to be warm, but not as warm as the areas of southwestern China that are influenced by the southwest Monsoon (Tan, 2007)". But when did it happen? MWP or LIA, or both?

Page 2 line 28, what does it mean "dry to wet to dry"?

Page 2 line 30, "Sever studies (Tan et al., 2003)." Incomplete sentence.

Page 4 line 15, "KY1 had uranium concentrations ranging from 704 to 5147 ppt". These are in fact thorium concentrations.

Page 4 line 18, figure 4 appears earlier than figure 3.

Page 5 line 18, not correct to have so many digits for isotope values. Same problem appears throughout the paper.

Page 7 line 16, "Comparing the $\delta$18O value curve to contemporaneous records of Swiss Alpine glaciation (Holzhauser et al., 2005) showed no obvious correlation (Fig. 7C)." Then what's the point to mention here?

Page 10 line 20, a clear definition is needed here for the "drought/waterlog index cumulative departure curve".

---

## Author Comment (AC1) · 20 Oct 2017

Author's response to anonymous Reviewer#1 - "Relationship between climate, environment, and anthropogenic activities in coastal North China recorded by speleothem ïĄď18O and ïĄď13C ratios in the last 1 ka)"

Qing Wang1, Ke Cheng1, Zhihui Zheng1, Hong Chi1, Hongyan Wang1 1Coast Institute of Ludong University, Yantai 264025, China Correspondence to: Qing Wang, schingwang@126.com; Ke Cheng, haian_cheng@126.com

[Figure]

Firstly, we would like to thank anonymous reviewer#1 for his/her comments and constructive suggestions, which will improve the manuscript, and for recommending this study for publication in Climate of the Past. Please find enclosed point by point responses to the comments. The referee suggestions and comments are displayed in red, and our answers in black.

General The manuscript entitled "Relationship between climate, environment, and anthropogenic activities in coastal North China recorded by speleothem ïĄď18O and ïĄď13C ratios in the last 1 ka" by Wang et al. extends the previous Kaiyuan cave record (Wang et al., 2015 Marine Geology and & Quaternary Geology; Wang et al., 2016 Clim. Past) from ∼AD1200 further back to AD 900. While the ïĄď13C record is new, the ïĄď18O is essentially the same as the data published previously. The majority of the discussions/conclusions is not only tentative and/or ambiguous (see examples as listed below), but also already published in Wang et al., 2015 Marine Geology and & Quaternary Geology and Wang et al., 2016 Clim. Past. As such, this manuscript is not suitable for considering publication in Climate of the Past.

In this manuscript, the stalagmite KY1 was dated by U-Th technique, and discussed the climatic-environmental meanings by ïĄď18O and ïĄď13C. The ïĄď18O ratios of the upper part of stalagmite KY1 has been discussed and published in Wang et al., 2015 Marine Geology and & Quaternary Geology; Wang et al., 2016 Clim. Past. As for this problem, the discussions of ïĄď18O ratios will be deleted substantially only in comparison with ïĄď13C ratios in the next modification. And the abstract will be improved.

Comments 1. More than half of the abstract is virtually as same as those in Wang et al., 2015 Marine Geology and & Quaternary Geology, and Wang et al., 2016 Clim. Past.

The abstract will be improved.

2. The link between the Kaiyuan record and Chinese cultural history is not convincing. For instance, if the Kaiyuan record is indeed a rainfall amount proxy on large spatial-scale in China, how about the differences with other existing records (such as Wangxiang, Heshang and Shihua records)? It really requires a detailed discussion how a record from 'the warm temperate zone (also need a definition)' can affect hydrological condition in China and thus the Chinese culture history.

Thank you for your comment. According to the record of stalagmite KY1, paleoclimate and history records, we discussed the correlation between the stalagmite record and the replacement of major dynasties of ancient China. We will find much more results and evidences to further research and verification.

3. It is necessary to give the reasoning why the 230Th age at $\sim$ 45mm was discarded.

As the laminae of the lower part is almost indistinguishable, we can't establish the timescale by the method of laminae counting.

4. The extended portion of the record has very poor age control and the methodology is problematic (e.g., the assumption of linear-growth is too weak). Thus, the new record cannot be used to address the issues in the way that presented in the current manuscript.

The dating results of the lower part of stalagmite KY1 is established by the methods of interpolation and extrapolation. By the boundary of the position of 64.5mm, we calculate the average growth rate of the part of 42.769mm-64.5mm first, and then extrapolate the age of the position of 75mm by the average growth rate. The position of 45mm is much close to the boundary of 42.769mm, so we chose position of 64.5mm. The expressions need to be improved.

5. The age uncertainties are not carefully considered throughout the manuscript when discussing relent issues such as age comparison, and the lead/lag among climate forcings. For example, the Five Dynasties and Ten Kingdoms has a duration less than the age uncertainty of the cave record at the time, and thus their correlation in the

Figure 5 needs a justification.

Thank you for your suggestion. The age uncertainties are determined by U-230Th technique, we will check and verify the discussion.

6. The authors interpreted the ïĄď13C record as an indicator of the land use. Given the fact of significant correlation between the ïĄď18O and ïĄď13C data (r=0.46, p<0.01), what about the ïĄď18O? Any anthropogenic (e.g., land use) effect? The data of land use are an overall summary from Shandong Province, which are not necessary to be equivalent to or describe the local variations at the cave site.

The ïĄď18O value of stalagmites in monsoonal areas like eastern China has been used as a proxy for the variability in the amount of rainfall because of the associated changes in the ïĄď18O values with changing moisture sources and shifting rainy season. Kaiyuan Cave is located at the warm temperate zone of the East Asia monsoon area. Rainfall is concentrated in the summer months, brought from the low latitudes of the Pacific by the summer monsoon. The data of land use of Shandong Province is alternative indicator to discuss the climatic-environmental meanings.

7. The authors had published the "Hendy Test" data already. In addition, the simple test presented in the manuscript is not necessary to be a robust verification of 'sample deposition under isotopic equilibrium'.

Thanks. This result has been published, we will check it. The expressions need to be improved.

8. The statement, "This report is the first example of a high-resolution study", is not proper, regarding many existing records, including that in authors' last paper (Wang et al., 2016 Clim. Past).

Thanks. We will delete this sentence.

9. The ïĄď18O variation is causally linked to the rainfall amount effect. This requires a very careful assessment.

The ïĄď18O value of stalagmites in monsoonal areas like eastern China has been used as a proxy for the variability in the amount of rainfall because of the associated changes in the ïĄď18O values with changing moisture sources and shifting rainy season. We will check the statements and discussions.

10. The reinterpretation of other cave records in the manuscript is problematic. For instance, the Wangxiang record is also an East Asian monsoon record, rather than a typical Westerlies record.

Wangxiang Cave is located in China's inland area, Kaiyuan Cave is located in coastal area. This comparison need to be improved.

11. Many climate records are now available for the last millennia in the East Asian monsoon region. I suggest considering a more comprehensive comparison. The comparison with records from Turkey and Europe is ambiguous and not helpful here, unless the authors provide a mechanism to explain their correlations.

Thank you for your comment. We are considering to compare with more achievements in East Asian monsoon region.

13. Almost all reported data in the manuscript have too many significant digits, which is obviously impossible.

All reported data in this manuscript are measured by professional equipment in laboratory, the sampling methods are expressed clearly in section 3. We are considering to improve the expressions.

14. The conclusion part is unusually long with many redundant contents.

Thanks. We will simplify the contents of conclusion.

15. The current manuscript is not sufficiently comprehensible, including English.

We will improve the expressions and consider to find language editing service by professional institution, and make the manuscript much easier to read.

16. Some references are not very appropriate and some need to update.

Thanks. We will check it.

17. Check the unit of U contents: ppt or ppb?

Thanks. The unit of U is ppb.

Please also note the supplement to this comment:
https://www.clim-past-discuss.net/cp-2017-73/cp-2017-73-AC1-supplement.pdf

---

## Author Comment (AC2) · 20 Oct 2017

Author's response to anonymous Reviewer#2 - "Relationship between climate, environment, and anthropogenic activities in coastal North China recorded by speleothem ïĄď18O and ïĄď13C ratios in the last 1 ka)"

Qing Wang1, Ke Cheng1, Zhihui Zheng1, Hong Chi1, Hongyan Wang1 1Coast Institute of Ludong University, Yantai 264025, China Correspondence to: Qing Wang, schingwang@126.com; Ke Cheng, haian_cheng@126.com

[Figure]

Firstly, we would like to thank anonymous reviewer#2 for his/her comments and constructive suggestions, which will improve the manuscript, and for recommending this study for publication in Climate of the Past. Please find enclosed point by point responses to the comments. The referee suggestions and comments are displayed in red, and our answers in black.

General

In their manuscript, Wang et al. tried to use oxygen and carbon stable isotope records in a stalagmite sample from Kaiyuan Cave to address the relationship between climate, environment and human activities. However, the manuscript was very poorly written and some parts are even unreadable (e.g., lines 25-30 in abstract and lines 23-25 in section 4.3). There are numerous grammar mistakes and redundancies in the writing (e.g., the first two paragraphs in section 3). Moreover, the authors often made awkward statements without reasoning. It is difficult for me to recommend its publication in Climate of the Past. Following I provide a few major comments and minor issues, and hope they would be helpful in authors' resubmission and future research.

Thank you for your comments. We will improve the expressions and consider to find language editing service by professional institution, and make the manuscript much easier to read.

Major issues

1. I am surprised that the current manuscript has a significant overlap with their previous paper published in Climate of the Past (Wang et al., 2016, 12, 871-881), although the authors did include new datasets (e.g., 2 more U/Th dates, ïĄď'13C data) in this manuscript. Some paragraphs were even simply copied from the previous one (e.g., in introduction, background, . . .). This is typically unacceptable in scientific journals.

In this manuscript, the stalagmite KY1 was dated by U-Th technique, and discussed the climatic-environmental meanings by ïĄď'18O and ïĄď'13C. The ïĄď'18O ratios of

the upper part of stalagmite KY1 has been discussed and published in Wang et al., 2016 Clim. Past. As for this problem, the discussions of ïĄď18O ratios will be deleted substantially only in comparison with ïĄď13C ratios in the next modification. All repeat contents will be improved.

2. The age model the authors constructed for the sample is not reliable. How could it be possible to assign a specific calendar year to a laminar, particularly given the slow growth rate and relatively large U/Th age error bar? Some U/Th dates were randomly thrown away, without careful reasoning.

The dating results of the upper part (0-42.769mm) of stalagmite KY1 has been published. The dating results of the lower part of stalagmite KY1 is established by the methods of interpolation and extrapolation. By the boundary of the position of 64.5mm, we calculate the average growth rate of the part of 42.769mm-64.5mm first, and then extrapolate the age of the position of 75mm by the average growth rate. The position of 45mm is much close to the boundary of 42.769mm, so we chose position of 64.5mm. The expressions need to be improved.

3. The authors really need to learn how to scientifically present their data. They need to add errors when present measurement data. Significant figures also need to be considered.

Thank you for your suggestion. We will check all the manuscript, and consider to modify the figures.

4. Section 4, it would be beneficial to show pictures of sample KY1, its lamina and micro-sampling locations.

We will add a picture to show the stalagmite and the sampling positions.

5. In section 4.1, why not re-measure the subsamples if the authenticity of the sample is uncertain?

The reason is technical limitation probably. We will check the error, and we will modify

the impressions.

6. In section 4.2, it seems very odd to exclude the U-Th age at 45mm from the age model.

The dating results of the lower part of stalagmite KY1 is established by the methods of interpolation and extrapolation. By the boundary of the position of 64.5mm, we calculate the average growth rate of the part of 42.769mm-64.5mm first, and then extrapolate the age of the position of 75mm by the average growth rate. The position of 45mm is much close to the boundary of 42.769mm, so we chose position of 64.5mm. The expressions need to be improved.

7. In section 4.3, the authors observe a quite interesting feature that "The UAT and So-fular Caves, affected by the maritime westerly wind, showed the latest ïĄď'18O muta-tion time; the Wanxiang and Spannagel Caves, nearly unaffected by the ocean, showed the earliest time; and the Kaiyuan Cave, affected by the East Asian Monsoon, fell in between." What's the possible mechanism behind this phenomenon?

Different climate types (e.g. monsoon area) and different location (e.g. inland or near the sea).

8. In section 4.3, the relationship between solar activity and ïĄď'18O were not suffi-ciently discussed. The authors argued changes in ïĄď'18O are corresponding to the variations in solar radiation (although I would argue it is not visually convincing). How-ever, the authors did not explain why the changes of ïĄď'18O lags the changes of solar radiation and why the changes of ïĄď'18O respond to solar changes differently before, during and after LIA. These statements are very subjective.

The relationship between solar activity and ïĄď'18O were not sufficiently discussed be-cause of limited space. The comparison need to be improved. We are also considering to delete this comparison.

9. In section 4.4, the authors argue that the variations of speleothem ïĄď'13C is controlled by the changes of proportion of C3- vs C4-, bio-productivity and the water-rock interaction. First, how to quantify the water-rock interaction? by time? Second, why the low ïĄď13C values before 1482AD necessarily "indicate a lower proportion of C3- versus C4-plants, lower bio-productivity, and less water-rock interaction" (page 9 line 10)? Third, "This behavior results in higher proportions of C3- versus C4-plants and increased bio-productivity." (page 10 line 33), which according to the authors will decrease the speleothem ïĄď13C values, contradicting to the increase trend of ïĄď13C values between 1480-1744 AD.

In this section, the quantification of the water-rock interaction is considered by time. The expressions need to be modified. The low ïĄď13C values before 1482AD was affected by the variations of natural vegetation. But the effects of anthropogenic activities were more and more prominent, and exceeded the natural factors finally.

10. In section 4.3 and 4.4, the relationship between the ancient Chinese dynasties and stable isotopes are very weak.

Thank you for your comment. According to the record of stalagmite KY1, paleoclimate and history records, we discussed the correlation between the stalagmite record and the replacement of major dynasties of ancient China. We will find much more results and evidences to further research and verification.

Minor issues Page 1 line 19, what does "smoother" mean here?

Thanks. This is a mistake. We will change this word to "more smooth".

Page 2 line 25, the authors stated "The areas of eastern and northern China influenced by the southeast monsoon are likely to be warm, but not as warm as the areas of southwestern China that are influenced by the southwest Monsoon (Tan, 2007)". But when did it happen? MWP or LIA, or both?

Thanks. We will add some expressions.

Page 2 line 28, what does it mean "dry to wet to dry"? Thanks. Dry to wet and back to

dry again. We will improve the expressions.

Page 2 line 30, "Sever studies (Tan et al., 2003)." Incomplete sentence.

Thanks. That's two references. We will correct this sentence.

Page 4 line 15, "KY1 had uranium concentrations ranging from 704 to 5147 ppt". These are in fact thorium concentrations.

Thanks. We will correct it.

Page 4 line 18, figure 4 appears earlier than figure 3.

Thanks. We will adjust the order.

Page 5 line 18, not correct to have so many digits for isotope values. Same problem appears throughout the paper.

OK. We will check the ïĄd'18O values.

Page 7 line 16, "Comparing the _18O value curve to contemporaneous records of Swiss Alpine glaciation (Holzhauser et al., 2005) showed no obvious correlation (Fig.7C)." Then what's the point to mention here?

Thanks. We are considering to delete this comparison.

Page 10 line 20, a clear definition is needed here for the "drought/waterlog index cumulative departure curve".

The cumulative departure curve of drought/waterlog index has been written in the published article (Wang et al., 2016, 12, 871-881) clearly. So we don't show it in this manuscript.

Please also note the supplement to this comment:
https://www.clim-past-discuss.net/cp-2017-73/cp-2017-73-AC2-supplement.pdf

---

## Author Comment (AC3) · 20 Oct 2017

Author's final response - "Relationship between climate, environment, and anthropogenic activities in coastal North China recorded by speleothem 18O and 13C ratios in the last 1 ka)"

Qing Wang1, Ke Cheng1, Zhihui Zheng1, Hong Chi1, Hongyan Wang1 1Coast Institute of Ludong University, Yantai 264025, China Correspondence to: Qing Wang, schingwang@126.com; Ke Cheng, haian_cheng@126.com

[Figure]

Firstly, we would like to thank two anonymous reviewers for their comments and constructive suggestions, which will improve the manuscript, and for recommending this study for publication in Climate of the Past.

According to the comments from the two anonymous reviewers, the main problem of this manuscript is the overlap with the previous paper published in Climate of the Past (Wang et al., 2016, 12, 871-881). In this paper, the stalagmite KY1 was dated by U-Th technique, and discussed the climatic-environmental meanings by 18O and 13C rations. The 18O ratios of the upper part of stalagmite KY1 has been discussed and published in Wang et al., 2016 Clim. Past. As for this problem, we have discussed many times about how to modify this manuscript. We decide to discuss the climatic-environmental meanings of 13C ratios primarily, the 18O ratios are comparison data. This manuscript has been modified. The main modifications are as follows.

1. The abstract and conclusion have been simplified. 2. We will add a picture to show the stalagmite and the sampling positions and adjust the order of all figures. 3. In section 4, we have deleted many discussions of 18O ratios, and only retain the comparison with 13C rations. We have also deleted the redundant figures. 4. Some of the references have been updated. We are still modifying this manuscript and checking the discussions and results based on the comments from the two anonymous reviewers. We will improve the expressions and consider to find language editing service by professional institution, and make the manuscript much easier to read. The whole manuscript which are modified up to now is in the supplement. Thank you for all the people who are concerned about this manuscript.

Regards, The authors (Qing Wang, Ke Cheng, Zhihui Zheng, Hong Chi, Hongyan Wang)  

Please also note the supplement to this comment:
https://www.clim-past-discuss.net/cp-2017-73/cp-2017-73-AC3-supplement.pdf

[Figure]

**Supplement:**

**Author's final response - "Relationship between climate, environment, and anthropogenic activities in coastal North China recorded by speleothem $\delta^{18}$O and $\delta^{13}$C ratios in the last 1 ka)"**

Qing Wang[1], Ke Cheng[1], Zhihui Zheng[1], Hong Chi[1], Hongyan Wang[1]

[1]*Coast Institute of Ludong University, Yantai 264025, China*

*Correspondence to*: *Qing Wang, schingwang@126.com; Ke Cheng,* haian_cheng@126.com

Firstly, we would like to thank two anonymous reviewers for their comments and constructive suggestions, which will improve the manuscript, and for recommending this study for publication in Climate of the Past.

According to the comments from the two anonymous reviewers, the main problem of this manuscript is the overlap with the previous paper published in Climate of the Past (Wang et al., 2016, 12, 871-881). In this paper, the stalagmite KY1 was dated by U-Th technique, and discussed the climatic-environmental meanings by $\delta^{18}$O and $\delta^{13}$C rations. The $\delta^{18}$O ratios of the upper part of stalagmite KY1 has been discussed and published in Wang et al., 2016 Clim. Past. As for this problem, we have discussed many times about how to modify this manuscript. We decide to discuss the climatic-environmental meanings of $\delta^{13}$C ratios primarily, the $\delta^{18}$O ratios are comparison data. This manuscript has been modified. The main modifications are as follows.

1. The abstract and conclusion have been simplified.
2. We will add a picture to show the stalagmite and the sampling positions and adjust the order of all figures.
3. In section 4, we have deleted many discussions of $\delta^{18}$O ratios, and only retain the comparison with $\delta^{13}$C rations. We have also deleted the redundant figures.
4. Some of the references have been updated.

We are still modifying this manuscript and checking the discussions and results based on the comments from the two anonymous reviewers. We will improve the expressions and consider to find language editing service by professional institution, and make the manuscript much easier to read.

The whole manuscript which are modified up to now is in the next page.

Thank you for all the people who are concerned about this manuscript.

Regards,

[revised manuscript text omitted]